# Cannabidiol and Oxygen-Ozone Combination Induce Cytotoxicity in Human Pancreatic Ductal Adenocarcinoma Cell Lines

**DOI:** 10.3390/cancers12102774

**Published:** 2020-09-27

**Authors:** Margherita Luongo, Oliviero Marinelli, Laura Zeppa, Cristina Aguzzi, Maria Beatrice Morelli, Consuelo Amantini, Andrea Frassineti, Marianne di Costanzo, Alessandro Fanelli, Giorgio Santoni, Massimo Nabissi

**Affiliations:** 1“Maria Guarino” Foundation—AMOR No Profit Association, 80078 Pozzuoli, Italy; margherita.luongo@aslnapoli2nord.it (M.L.); info@dottorfrassineti.it (A.F.); marianne.dicostanzo@ospedalideicolli.it (M.d.C.); 2School of Pharmacy, University of Camerino, 62032 Camerino, MC, Italy; oliviero.marinelli@unicam.it (O.M.); laura.zeppa@studenti.unicam.it (L.Z.); cristina.aguzzi@studenti.unicam.it (C.A.); mariabeatrice.morelli@unicam.it (M.B.M.); giorgio.santoni@unicam.it (G.S.); 3School of Bioscience and Veterinary Medicine, University of Camerino, 62032 Camerino, MC, Italy; consuelo.amantini@unicam.it; 4Department of Radiotherapy, Ecomedica Empoli, 50053 Empoli, FI, Italy; alessandro.fanelli@ecomedica.it; 5Integrative Therapy Discovery Lab, University of Camerino, 62032 Camerino, MC, Italy

**Keywords:** cannabidiol, oxygen-ozone, pancreatic cancer, cytotoxicity, migration, chemo-resistance, PDAC gene profile

## Abstract

**Simple Summary:**

Pancreatic cancer (PC) is related to lifestyle risks, chronic inflammation, and germline mutations. Surgical resection and adjuvant chemotherapy are the main therapeutic strategies but are less effective in patients with high-grade tumors. Oxygen-ozone (O_2_/O_3_) therapy is an emerging alternative tool for the treatment of several clinical disorders. The advantages of using cannabinoids have been evaluated in several human cancers. Regarding PC, activation of cannabinoid receptors was found to induce PC cell apoptosis without affecting the normal pancreas cells. Herein, we evaluate the anticancer effect of cannabidiol (CBD) and O_2_/O_3_, alone or in combination, on two human pancreatic ductal adenocarcinoma (PDAC) cell lines, PANC-1 and MiaPaCa-2, examining expression profiles of 92 pancreatic adenocarcinoma associated genes, cytotoxicity, migration properties, and cell death. Finally, we assess the combination effects with gemcitabine and paclitaxel. Summarizing, for the first time the antitumoral effect of combined therapy with CBD and oxygen-ozone therapy in PDAC is evidenced.

**Abstract:**

Pancreatic cancer (PC) is related to lifestyle risks, chronic inflammation, and germline mutations in *BRCA1/2*, *ATM*, *MLH1*, *TP53*, or *CDKN2A*. Surgical resection and adjuvant chemotherapy are the main therapeutic strategies but are less effective in patients with high-grade tumors. Oxygen-ozone (O_2_/O_3_) therapy is an emerging alternative tool for the treatment of several clinical disorders. O_2_/O_3_ therapy has been found to ameliorate mechanisms promoting chronic pain and inflammation, including hypoxia, inflammatory mediators, and infection. The advantages of using cannabinoids have been evaluated in vitro and in vivo models of several human cancers. Regarding PDAC, activation of cannabinoid receptors was found to induce pancreatic cancer cell apoptosis without affecting the normal pancreas cells. In a murine model of PDAC, a combination of cannabidiol (CBD) and gemcitabine increased survival length by nearly three times. Herein, we evaluate the anticancer effect of CBD and O_2_/O_3_, alone or in combination, on two human PDAC cell lines, PANC-1 and MiaPaCa-2, examining expression profiles of 92 pancreatic adenocarcinoma associated genes, cytotoxicity, migration properties, and cell death. Finally, we assess the combination effects with gemcitabine and paclitaxel. Summarizing, for the first time the antitumoral effect of combined therapy with CBD and oxygen-ozone therapy in PDAC is evidenced.

## 1. Introduction

Pancreatic cancer (PC) is a lethal malignancy with a 5-year survival of approximately 5–9% [1,2]. The most common and aggressive type, among pancreatic malignancies, is pancreatic ductal adenocarcinoma (PDAC), an infiltrating neoplasm with glandular differentiation, that is the fourth cause of cancer related death worldwide [2,3,4]. The development of PDAC is related to environmental and lifestyle risks, but also pathological conditions linked to chronical inflammations and, for a subgroup of PDAC patients, germline mutations in Breast Cancer Type 1/2 (*BRCA1/2*), ATM Serine/Threonine Kinase (*ATM*), MutL Homolog 1 gene (*MLH1*), *TP53* or Cyclin Dependent Kinase Inhibitor 2A (*CDKN2A*) are considered further risk factors [5,6]. Indeed, somatic mutations as in oncogene (*KRAS*) and onco-suppressor genes (*TP53*, *CDKN2A*, SMAD Family Member 4 *SMAD4*), that support cancer aggressiveness, are important for the diagnosis of this malignant phenotype [2,7]. Pancreatic cancer hardly responds to chemotherapy or immunotherapy, and surgery is usually not an option in most patients [8]. Indeed, surgical resection followed by adjuvant chemotherapy is the main therapeutic strategy for the 10–20% of patients with resectable PDAC stage, but is less effective in the rest of patients that show locally advanced, non-resectable stages or distant metastasis [9,10]. For these non-resectable patients, systemic chemotherapy is employed as first-line treatment, with different drugs as gemcitabine (GEM), capecitabine, 5-fluorouracil in monotherapy or in combination with radiotherapy [11]. New potential therapies such as immunotherapy based on checkpoint inhibitors and targeted therapy are under evaluation but actually these treatments showed limited efficacy and were unsuccessful in improving patients’ survival, especially for metastastic pancreatic cancer [4,12,13]. These difficulties in treating PDAC are probably due to the special tumor-associated microenvironment. Unlike most other solid cancers, the pancreatic cancer microenvironment contains ample stromal cells, lacks vascularization, and cancer cells survive in long-term hypoxic conditions via special, but not completely identified, mechanisms [14]. Hypoxia has been reported to induce aggressive characteristics in many types of cancer, modulating different genes involved in cell proliferation, drug efflux, apoptosis, metabolism, autophagy, and angiogenesis, contributing to drug resistance [14,15]. Under hypoxia, tumor cells develop various mechanisms for evading apoptosis as over-expression of Mouse Double Minute Homolog (*MDM2*), the negative regulator of p53, leading to apoptosis resistance [16]. Moreover, hypoxic cells do not receive enough chemotherapeutic agents because they are distant from the capillary and because of abnormal vascularization of tumors [16]. Therefore, tumor hypoxia has been considered as a validated target for treating cancer [17,18].

Oxygen/Ozone (O_2_/O_3_) therapy is an emerging alternative tool for treatment of several clinical disorders, among them, ischemic disorder and tuberculosis. In the past, the use of O_2_/O_3_ in therapy was discouraged by orthodox medicine for the lack of solid scientific biological and clinical data [19]. Recently, some concerns have been practically overcome both at in vitro and in vivo level. O_2_/O_3_ has shown anti-inflammatory effects [20] and the improvement of muscle fatigue [21]. O_2_/O_3_ is a gas characterized by three atoms of oxygen with a cyclic structure and the medical generator of ozone produces it from pure oxygen passing through a high voltage gradient (5–13 mV). One of the most important properties of O_2_/O_3_ administration is the increase in tissue oxygenation and ozone therapy, which has been found to ameliorate many of the mechanisms promoting chronic pain and inflammation, including hypoxia, inflammatory mediators, and infection [22]. In cancer, O_2_/O_3_ was found to inhibit the growth of different human tumor cells (breast, colon, ovarian) without affecting non-tumor cell lines and to potentiate the effect of chemotherapeutic drugs as 5-fluorouracil, cisplatin, and etoposide [23,24]. Advantages of using cannabinoids in cancer therapy have been evaluated in in vitro and in vivo models of several human cancers. Their potential use for cancer in current clinical trials has been recently reviewed [25]. Indeed, cannabinoids can reduce cancer cell viability, proliferation, metastasis, and they induce cancer cell death in human glioblastoma multiforme, multiple myeloma, colorectal cancer, endometrial cancer, prostate carcinoma, and melanoma [26,27,28,29]. Cannabidiol (CBD), the main non-psychotropic cannabinoid, has been demonstrated to interact with cannabinoid receptors (CB1, CB2), G Protein-Coupled Receptor 55 (GPR55), with transient potential receptors belonging to vanilloid subfamily (TRPV1, TRPV2, TRPV3, TRPV4), Transient Receptor Potential Ankyrin 1 (TRPA1), Transient Receptor Potential Melastatin 8 (TRPM8), with peroxisome proliferator-activated receptor (PPARγ), but it can also act in an unknown independent-receptors manner [26,30]. Regarding PDAC, activation of cannabinoid receptors, particularly CB2, was found to induce pancreatic cancer cell apoptosis without affecting the normal pancreas cells [31] and different synthetic receptor agonists, as WIN-55,212-2 (CB1 and CB2), ACEA (CB1), and JWH-015 (CB2), caused a substantial cell death of MiaPaCa-2 cell [32]. In vivo study demonstrated that a combination of CBD (as GPR55 antagonist) and GEM in murine model of pancreatic cancer, increased a survival nearly three times longer, compared to mice treated with vehicle or GEM alone [33]. Similar findings were shown in a study with the CB1 ligand SR141716 combined with GEM, where was evidenced a reduced tumor growth in treated mice with respect to control and single agent treated mice [34]. As to O_2_/O_3_, CBD exerts a synergistic activity with different chemotherapeutic drugs [25,26]. So, the aim of this research was to investigate, for the first time, the anticancer effect of CBD and O_2_/O_3_, alone and in combination, on two human PDAC cell lines, PANC-1 and MiaPaCa-2, by examining expression profiles of 92 pancreatic adenocarcinoma associated genes, cytotoxicity, migration properties, and cell death. Finally, the effects of the combination with GEM and Paclitaxel (PTX), the main chemotherapeutical drugs used in pancreatic cancer therapy, were assessed.

## 2. Results

### 2.1. Expression of CBD Target Receptors in PDAC Cell Lines

The gene and protein expressions of CB1, CB2, TRPV1, TRPV2, TRPV3, TRPV4, TRPA1, and TRPM8 were evaluated in PANC-1 and MiaPaCa-2 cancer cell lines by qRT/PCR and Western Blot analysis. Overall, gene and protein expression showed that CB1 and CB2 receptors are expressed in both cell lines without any significant difference for the CB1 receptor, while the CB2 protein level is higher in the PANC-1 cell line. Regarding the expression profile of the TRP members analyzed, PDAC cells express all CBD targets belonging to this family, both at mRNA and at protein levels. Moreover, TRPV3 and TRPV4 proteins are highly expressed in both PDAC cell lines, especially in MiaPaCa-2 cells, and TRPV2 is expressed mainly in the PANC-1 cell line. Additionally, at protein levels, TRPM8 is expressed more in PANC-1 cells while TRPA1 is not expressed in both PDAC cell lines (Appendix A). Summarizing, PDAC cells express significant levels of most of the CBD target receptors and the difference observed between PANC-1 and MiaPaCa-2 cell lines could suggest a different response to CBD.

### 2.2. CBD is More Effective in Reducing Cell Viability in PDAC Cell Lines than in Normal Cells

The cytotoxic effect of CBD was evaluated in PANC-1 and MiaPaCa-2 cancer cell lines and in two normal cell lines H6c7 and NHF, by 3-[4,5-dimethylthiazol-2-yl]-2,5 diphenyl tetrazolium bromide (MTT) assay. Cells were treated with different doses of CBD (from 1.52 to 100 μM) with daily administration up to 72 h. The results showed that cell viability is reduced in a dose-dependent manner with an IC_50_ of 20.3 ± 0.4 μM for PANC-1 and an IC_50_ of 18.6 ± 1.2 μM for MiaPaCa-2 (Figure 1), indicating a major sensitivity of MiaPaCa-2 with respect to PANC-1, while H6c7 and NHF cell line results showed less sensitivity to CBD (Appendix A). According to IC_50_ values, we decided to use CBD at 12.5 and 25 μM, in daily administration, for the next experiments.

### 2.3. CBD Induces Apoptotic Cell Death in PDAC Cancer Cell Lines

To assess cell death, FITC-conjugated Annexin V and Propidium Iodide (PI) staining and cytofluorimetric analysis were used. After 48 h of daily treatment with CBD (12.5–25 μM), it was observed that CBD induces an increased percentage of cells undergoing apoptosis compared to control, in both cell lines. PANC-1 and MiaPaCa-2 showed a significant increase in apoptotic cell death with CBD 25 μM compared to 12.5 μM (Figure 2).

To confirm apoptosis, Caspase 3 (Casp3) activation was evaluated, by Western Blot analysis. Cells were treated with CBD 25 μM for 48 h in daily administration and the results confirm an increase in activated Casp3 in both cell lines, especially in MiaPaCa-2 cells (Figure 3A). Moreover, by Comet assay analysis, we confirmed that the CBD 25 μM after 48 h of treatment induced DNA damage (Figure 3B).

### 2.4. CBD Reduces Cell Migration of PDAC Cell Lines

To examine the role of CBD in regulating migration of PANC-1 and MiaPaCa-2 cells, the wound-healing assay was performed. The results showed that CBD 12.5 μM, does not induce a significant effect in cell migration after 24 h of treatment, while at 48 h, a reduction of cell migration is observed in both cell lines (Figure 4). These data suggested that CBD influences PDAC cell line migration.

### 2.5. CBD Increases Chemosensitivity in PDAC Cell Lines

In order to evaluate a synergistic effect between CBD and the most common chemotherapeutic drugs used in PDAC treatment, GEM (up to 800 µM) and PTX (up to 28 µM) were tested in both cell lines. The results evidenced that MiaPaCa-2 cells are more sensitive to GEM and PTX than PANC-1 and that PTX shows a higher cytotoxic effect than GEM, in both cell lines (PANC-1 GEM IC_50_: 143.8 ± 2.4 μM; PTX 33.57 ± 1.2 nM, MiaPaCa-2 GEM IC_50_: 63.6 ± 3.5 μM; PTX 21.18 ± 1.1 nM) (Figure 5), at 72 h post-treatments.

Subsequently, PDAC cell lines were exposed to CBD at 6.25, 12.5, and 25 μM in combination with three doses of each chemotherapeutic drug (GEM 100, 50, and 25 μM, and PTX 7, 3.5, and 1.75 µM) for 72 h. Chemotherapeutic drugs were administered once, while CBD was administered daily. The results showed that CBD 6.25 μM, in combination with all tested doses of GEM and PTX, did not increased cytotoxic effects compared with chemotherapeutic drugs alone, while CBD 12.5 and 25 μM was able to increase the cytotoxic effect induced by both chemotherapeutic drugs alone (Figure 6). Indeed, synergistic effects were obtained with PTX 7 μM especially in MiaPaCa-2 cells while, in PANC-1, this combination resulted in an additive effect. Combination with GEM 100 μM induced an additive and synergistic effect with, respectively, CBD 12.5 and 25 μM, in MiaPaCa-2 but not in the PANC-1 cell line (Appendix A). The results evidenced that CBD, at the appropriate doses, should be useful to enhance the chemotherapeutic drugs effects.

### 2.6. O_2_/O_3_ Treatment Improves CBD Cell Cytotoxicity in PDAC Cells

The effect of O_2_/O_3_ on PDAC cell lines was evaluated by using a hypoxia chamber. PANC-1 and MiaPaCa-2 cell lines were daily treated with O_2_/O_3_ and cell viability, with respect to cells cultured in normoxia, was evaluated up to 72 h, by MTT. Additionally, the effect of CBD administration in O_2_/O_3_-treated cell lines was evaluated. The results showed that the addition of O_2_/O_3_ strongly reduces cell viability, as observed starting from 24 h post-treatment, and the effect of O_2_/O_3_ was increased by CBD in a dose and time dependent manner, in both cell lines (Figure 7).

Furthermore, H6c7 and NHF were daily treated with CBD and O_2_/O_3_, as performed with PANC-1 and MiaPaCa-2 cell lines, and cell viability was evaluated at 72 h post-treatment. The results evidenced that O_2_/O_3_ did not influence H6c7 and NHF cell viability and did not increase the CBD-effects (Appendix A). Summarizing, H6c7 and NHF cell lines resulted less sensitive to CBD and O_2_/O_3_ both administered alone and in combination, as described in Table 1.

Moreover, cell death was examined at 24 h post-treatment, by Annexin V staining and PI incorporation assay and results showed that O_2_/O_3_ is able to induce necrotic cell death in both cell lines, and that O_2_/O_3_-CBD treatment increases the percentage of PI positive cells (Figure 8).

To further evaluate the role of O_2_/O_3_ in modulating the effect of CBD in combination with chemotherapeutic drugs, both cell lines were treated with GEM or PTX up to 48 h after O_2_/O_3_ addition. Cell viability was calculated comparing with untreated cells in presence of O_2_/O_3_. The results evidenced that the combination of O_2_/O_3_ with CBD plus chemotherapeutic drugs increases the cytotoxic activity showed by O_2_/O_3_ alone and by CBD plus chemotherapeutic drugs without O_2_/O_3_, as shown (Figure 9). Summarising, these results indicate that the combination O_2_/O_3_ plus CBD improves the cytotoxic activity of chemotherapeutic drugs.

### 2.7. Gene Modulation by CBD and O_2_/O_3_ Treatments in PDAC Cell Lines

To elucidate the molecular events induced by CBD and O_2_/O_3_ treatment, 92 genes involved in PDAC progression and aggressiveness were evaluated by Taqman Array. Both cell lines were treated with CBD, O_2_/O_3_, and with CBD plus O_2_/O_3_, and the molecular pathways of PDAC associated gene were evaluated. As shown, different pathways involved in PDAC carcinogenesis were modulated by CBD, O_2_/O_3_, and by their combination, suggesting that both treatments influence common but also specific pathways. Regarding the effect in regulating genes involved in cell cycle progression, the major effects of CBD, after 24 h of treatment, was in down-regulation of Cyclin A2 (*CCNA2*) (Figure 10A, Appendix A). The effect of O_2_/O_3_ should be considered as very effective in regulating cell cycle since all the Cyclins (*CCNs*), Cyclin Dependent Kinases (*CDKs*) and E2Fs (*E2F1*, *E2F3*) analysed were down-regulated after 24 h of treatments, while the *CDKN2A* expression was up-regulated, compared to vehicle-treated cells (Figure 10B, Appendix A). Moreover, the co-treatments with O_2_/O_3_ and CBD, confirmed the effect of O_2_/O_3_ and, additionally, further up-regulated *CDKN2A* with respect to O_2_/O_3_-treated cells (Figure 10B, Appendix A). To support the modulation of cell cycle gene pathways by CBD and O_2_/O_3_, PI staining and cell cycle analysis was performed. As reported, CBD and O_2_/O_3_ alone slightly increased G2/M phase while the combination of both induced also a slight rise in Sub-G1 phase, in both cell lines (Appendix A).

Summarising, these data suggest that the co-treatment with O_2_/O_3_ and CBD was efficient in reducing cell cycle progression in PDAC cell lines, regulating several markers associated with cancer cell proliferation.

Moreover, we found that CBD significantly reduced ETS Like-1 protein Elk-1 (*ELK1*), Erb-B2 Receptor Tyrosine Kinase 2 (*ERBB2*), Mitogen-activated Protein Kinase Kinase 1 (*MAP2K1*), and *RAF-1*, while no effects were observed in Ras-pathways inhibitors, compared to vehicle treated cells (Figure 11, Appendix A). Comparing O_2_/O_3_ treatments with vehicle-treated cells, we found that all the Ras-associated pathways genes were down-regulated, while no effect was observed for Rac Family Small GTPase 2 (*RAC2*) (Figure 11, Appendix A). The combination with CBD induced a total inhibition of *BRAF* gene expression and strongly increased *MAP2K1*, *MAP2K2*, and *ERBB2* gene levels compared with O_2_/O_3_ alone. Furthermore, no significant additional effects on the others Ras-associated genes were observed (Figure 11, Appendix A).

Thanks to Taqman Array we also detected that CBD significantly increases the *TP53* gene, which is associated with DNA repair (Figure 12, Appendix A). O_2_/O_3_ treatment was effective in increasing *BRCA2* and *TP53* gene expression with respect to vehicle-treated cells (Figure 12, Appendix A). However, the combination with CBD reduced *BRCA2* and *TP53* gene expression, restoring the expression of untreated cells (Figure 12, Appendix A).

Regarding the Nf-kB pathway, CBD significantly increased *RELB* (Figure 13, Appendix A). O_2_/O_3_ treatment was effective in reducing *NF-kB2*, *REL* and Ras Homolog Family Member A (*RHOA*) gene expression with respect to vehicle-treated cells (Figure 13, Appendix A). The combination with CBD did not produce additional effects, except a further reduction in *RHOA* gene level. (Figure 13, Appendix A).

Regarding Phosphatidylinositol 3-kinase/Protein kinase B (PI3K/AKT) pathway, CBD did not significantly modulate the expression of the main genes. Indeed, it induced a slight increase in *PIK3CD* and only *PIK3CB*, *PIK3R1* and *PIK3R2* were reduced. (Figure 14, Appendix A). O_2_/O_3_ treatments reduced the expression of all analyzed genes compared with vehicle-treated cells (Figure 14B, Appendix A), and the combination with CBD maintained the effect of O_2_/O_3_ (Figure 14, Appendix A).

## 3. Discussion

PDAC patients present for 80–90% non-resectable stage cancer or distal metastasis, so systemic chemotherapy is applied as first-line treatment [10,35]. This therapy includes GEM, PTX, and nucleoside analogues [36] in monotherapy or in combination with radiotherapy or by a poly-chemotherapeutic regiment [37]. However, so far treatment efficacy in PDAC is considered limited. The interest associated with cannabinoids administration is related to their palliative effects useful for the treatment of cancer pain, but also for their activity as anticancer compounds able to induce inhibition of cancer cell growth and increasing cancer cell death. CBD anticancer effects, recently reviewed, were investigated in in vitro and in vivo models of glioma, leukaemia, breast, lung, thyroid, colon carcinoma, myeloma, and melanoma [25]. CBD activities are mediated through the binding to different receptors (CB1, CB2, and Gpr55 receptors), and some members of TRP channels family (as TRPV1-2-3-4, TRPM8 and TRPA1), but also by a receptor-independent mechanism [38,39]. We firstly profiled CBD-ligand receptors expression, evidencing low levels of CB1 in both cell lines, while CB2 receptor was more expressed in PANC-1 than in MiaPaCa-2. Previously, CB1 and CB2 expression was evaluated in four human pancreatic cancer cell lines and biopsies compared to normal pancreatic biopsies. Data showed different CB1 and CB2 expression levels in pancreatic cancer cell lines, compared to normal pancreatic tissues [40]. Moreover, according to our results, mRNA levels of CB1 and CB2 showed equal quantities of CB1 receptor in PANC-1 and MiaPaCa-2 cell lines and confirmed higher levels of CB2 receptor in PANC-1 respect to MiaPaCa-2 [41].

Up to now, the role of CBD in inducing anticancer effect in pancreatic cancer has not been well characterized. However, it was found that ∆9-tetrahydrocannabinol (THC) decreased PANC-1 and MiaPaCa-2 cell viability and induced apoptotic cell death both in vitro and in vivo [40]. Moreover, CBD through a GPR55 receptor antagonism was evidenced to reduce cell growth, cell cycle progression, and Mitogen-Activated Protein Kinase 1 (MAPK) signaling in different PDAC cell lines and the combination of CBD and GEM, was more effective compared with either treatment alone [33,34]. GEM and PTX are chemotherapeutic drugs normally used for pancreatic cancer therapy. In our study the combined treatment with CBD/GEM and CBD/PTX was able to reduce cancer cells viability, and showed an increased effect underlined using different doses of CBD and chemotherapeutic drugs. Our study also showed CBD’s ability to reduce PANC-1 and MiaPaCa-2 migration and similar evidence was found in other cancer cell lines [26,42].

Regarding O_2_/O_3_ therapy, evidence has been provided of the anti-cancer effect of local administration of ozonated water treatment in vitro and in vivo, in a mouse model of rectal cancer. The local treatment induced damages only in the tumor tissues by inducing necrosis without affecting normal tissues [43]. Moreover, the inhibitory effect of O_2_/O_3_ treatment in two human neuroblastoma cell lines was analyzed [24]. In this study, it was demonstrated that O_2_/O_3_ treatment was able to reduce cell growth and to arrest cell cycle at G2 phase, by inhibiting the expression and localization of cyclin B1/cdk1 in neuroblastoma cell lines. Additionally, it was also evidenced that O_2_/O_3_ improved the responsiveness to Cisplatin, Etoposide, and GEM [24]. These results support our evidence in pancreatic cell lines in which O_2_/O_3_ administration strongly reduced PDAC cells viability, inducing necrotic cell death, and it was shown that O_2_/O_3_ treatment is able to act synergistically with the common chemotherapeutic drugs used in pancreatic cancer management, PTX and GEM. Thus, supported by these results, we focused our attention on CBD and O_2_/O_3_ combination, since no data were available about this potential co-administration. We found that CBD plus O_2_/O_3_ enhanced the effect of both single treatments in reducing cell viability and increasing cell death, with effects observed just after 24 h of co-treatments, suggesting that the early effect was predominantly attributable to O_2_/O_3_. CBD was previously reported to have less cytotoxicity in non-tumoral cells [44]. The process that regulates the lower effects of O_2_/O_3_ in non-tumoral cells is not well investigated. However, different research papers evidenced that O_2_/O_3_ has a protective effect in normal cells, while it is toxic for cancer cells [45,46].

Different whole-exome sequencing studies on PDAC elucidated the major mutations and somatic copy number alterations including *KRAS*, *TP53*, *SMAD4*, *CDKN2A*, and damage repair genes such as *BRCA1*, *BRCA2*, giving rise to high cell proliferation and migration, reduction of cell death and genomic instability [47,48,49]. Since very little information relating to the molecular pathways involved in CBD and O_2_/O_3_ effects are available, we investigated which genes could be modulated in PDAC cell lines, by analyzing 92 PDAC associated genes. Regarding cell cycle and proliferation pathways, we found that CBD reduced the expression of *CCNA2*. O_2_/O_3_ strongly inhibited most of the genes that stimulate cell cycle—all of the *CCNs*, *CDKs*, and *E2Fs* (*E2F1*, *E2F3*)—while *CDKN2A* expression was up-regulated. In addition, the co-treatments with O_2_/O_3_ and CBD further up-regulated *CDKN2A* with respect to O_2_/O_3_-treated cells. Similarly, CBD was found to inhibit Ras-associated genes such as *ERBB2*, *RAF-1*, *ELK1*, *MAP2K1* and these effects were maintained in co-treatments with O_2_/O_3_ for *ELK1* and *RAF-1*, evidencing another potential mechanism to reduce PDAC gene profile. These data suggested that CBD and O_2_/O_3_ should have an important role in dis-regulating the Ras pathways and its downstream CDKs signaling, and that could have potential therapeutic implications, since this pathway is correlated with various oncogenic signals, such as proliferation, chemoresistance, and migration in PDAC, as recently reviewed [50]. In addition, it should be taken into consideration that pre-clinical data revealed pharmacological inhibition of Ras pathways and MAPK signaling results in compensatory activation of PI3K/Akt signaling and vice versa, both of them necessary for PDAC progression [51,52] and that independent of Ras pathways, *PI3K*/*Akt* is often elevated in PDAC and correlated to tumor cell survival [53,54]. Herein, we also evidenced that CBD and O_2_/O_3_, as single treatment or in combination, are able to reduce the expression of different genes of the PI3K pathway. PDAC, as above described, is also characterized by the acquisition of an anti-apoptotic phenotype, partially by inhibition of p53 and over-expression of anti-apoptotic proteins, which correlate with short survival and overall survival, particularly in Bcl-2 positive cases [55,56], and mutation in genes related to DNA damage response, such as *BRCA1/2* [54,57]. Herein, we showed that CBD alone increased *p53* and reduced *BRCA2* gene expression. Moreover, the combination with O_2_/O_3_ restored the expression of untreated cells. In different cancer models, it was demonstrated that BRCA2 leads to accumulation of DNA breaks, and results in activation of p53, which promotes cell cycle arrest and activation of cell death [58,59]. Moreover, in cancer, BRCA2 inactivation leads to pro-inflammatory cytokines production, that is a determinant for cancer cell survival [60], and several studies have investigated the expression profile of various cytokines in patients with PDAC and Nuclear Factor kB (NF-κB) activation pathways that have also been shown to be involved in pancreatic cancer development [61,62,63]. Moreover, inhibiting NF-κB and its downstream targets, lead to the inhibition of proliferation, angiogenesis, and invasion as reported in the PDAC mouse model [64]. Herein, the effects of CBD and O_2_/O_3_ on Nf-kB pathways were analyzed, and the results evidenced that O_2_/O_3_ alone and in combination with CBD was able to reduce NF-kB-related genes supporting their role in reducing inflammation and potentially cellular migration.

Summarizing, our results showed that CBD and O_2_/O_3_ were both able to induce significant changes in the expression profile of genes strongly involved in PDAC leading to the inhibition of cell viability, invasion, and increasing cell death. Moreover, CBD and O_2_/O_3_ was found to increase the anti-tumoral effects of Gemcitabine and Paclitaxel, suggesting that these combinations could have significant potential as an effective therapy for pancreatic cancer that can enhance the effect of chemotherapy and overcome chemoresistance.

## 4. Materials and Methods

### 4.1. Cell Lines

Human pancreatic ductal adenocarcinoma (PANC-1 and MiaPaCa-2) cell lines were purchased by Sigma Aldrich (Milan, Italy) and cultured in DMEM high glucose medium (EuroClone, Milan, Italy) supplemented with 10% of fetal bovine serum (FBS), 2 mM L-glutamine, 100 IU/mL penicillin, 100 mg streptomycin and 1 mM sodium pyruvate. Human Pancreatic Duct Epithelial H6c7 cell line was purchased by Kerafast (Boston, MA, USA) and cultured in Keratinocyte serum free medium, supplemented with epidermal growth factor and bovine pituitary extract. Normal Human Fibroblasts NHF cell line was purchased by IFOM (Rome, Italy) and cultured in DMEM supplemented with 10% of fetal bovine serum (FBS), 2 mM L-glutamine, 100 IU/mL penicillin, 100 mg streptomycin and 1 mM sodium pyruvate. Cell lines were maintained at 37 °C with 5% CO_2_ and 95% of humidity.

### 4.2. RNA Isolation, Reverse Transcription and Quantitative Real-Time PCR and TaqMan Array

Briefly, total RNA from untreated or CBD and O_2_/O_3_ treated cell lines was extracted using Rneasy Mini kit (Qiagen, Milan, Italy). One microgram of total RNA from each sample was subjected to reverse transcription in a total volume of 20 µL using the High-Capacity cDNA Archive Kit (Applied Biosystem, Foster City, PA, USA) according to the instructions. cDNAs were analyzed by qRT-PCR performed using an IQ5 Multicolor Real time PCR Detection system. Quantitative real-time polymerase chain reactions (qRT-PCR) were performed with QuantiTect Primer Assays (Qiagen) for Human Cannabinoid receptor 1 (*CNR1*, *CB1*), Human Cannabinoid receptor 2 (*CNR2*, *CB2*) and TRP (*TRPV1*, *TRPV2*, *TRPV3*, *TRPV4*, *TRPM8*, *TRPA1*), according to manufacturer’s protocol. Measurement of *GAPDH* levels were used to normalize mRNA contents and target gene levels were calculated by 2^−ΔΔCt^ method. The TaqMan**^®^** Array Human Pancreatic Adenocarcinoma 96-well Plate, containing 92 assays to pancreatic adenocarcinoma associated genes and 4 assays to candidate endogenous control genes, was purchased (Thermo Fisher, Grand Island, NY, USA) and used to evaluate the effects of the treatments in modulating PDAC-related genes. Measurement of two housekeeping genes (*GAPDH*; *HPRT1*; *GUSB*) on the samples was used to normalize mRNA content. The gene expression levels of treated cell lines were expressed as relative fold compared with untreated or vehicle-treated cells [65].
(1)$$ratio=\frac{{\left(E_{target}\right)}^{\Delta CP_{target}\left(control-sample\right)}}{{\left(E_{ref}\right)}^{\Delta CP_{ref}\left(control-sample\right)}}$$


### 4.3. Western Blot Analysis

Lysates of PANC-1 and MiaPaCa-2 cell lines untreated or daily treated with CBD for 48 h, were obtained with lysis buffer (composed by TRIS 1M pH 7.4, NaCl 1M, EGTA 10 mM, NaF 100 mM, Deoxycholate 2%, EDTA 100 mM, TritonX-100 10%, Glycerol, SDS 10%, Na_2_P_2_O_7_ 1M, Na_3_VO_4_ 100 mM, PMSF 100 mM, Cocktail of enzyme inhibitors and H_2_O). Lysates were separated on a SDS polyacrylamide gel, transferred onto Hybond-C extra membranes (GE Healthcare, Chicago, IL, USA), blocked with 5% low-fat dry milk in phosphate-buffered saline 0.1% Tween 20 overnight at 4 °C, immunoblotted with mouse anti-CB1 (1:500, Santa Cruz Biotechnology, Heidelberg, Germany), rabbit anti-CB2 (1:200, Cayman Chemical, Ellsworth, MI, USA), mouse anti-TRPV1 (1:200, Santa Cruz Biotechnology), mouse anti-TRPV2 (1:200, Santa Cruz Biotechnology), rabbit anti-TRPV3 (0.5 µg/mL, Boster Biological Technology, Pleasanton, CA, USA), rabbit anti-TRPV4 (1:500, Assay Biotechnology Company, Fremont, CA, USA), goat anti-TRPA1 (1:300, Santa Cruz Biotechnology), rabbit anti-TRPM8 (0.5 µg/mL, Boster Biological Technology), mouse anti-glyceraldehydes-3-phosphate dehydrogenase (GAPDH, 1:3000, OriGene, Rockville, MD, USA), and rabbit anti-caspase-3 (1:1000, Cell Signaling, Danvers, MA, USA) Abs overnight or 1 h according to manufacturer’s protocol and then incubated with their respective HRP-cojugated anti-rabbit, anti-mouse (1:2000, Cell Signaling, Danvers, MA, USA) or anti-goat (1:1000, Santa Cruz Biotechnology) Abs for 1 h. Peroxidase activity was visualized with the LiteAblot**^®^**PLUS or TURBO (EuroClone, Milan, Italy) kit and densitometric analysis was carried out by a Chemidoc using the Quantity One software version 4.6 (Bio-Rad, Milan, Italy). All experiments were repeated three times.

### 4.4. Reagents

Pharmaceutical grade Cannabidiol (CBD) crystals were purchased (Pharmacy S. Albano, Torino, Italy). CBD crystals were solubilized in ethanol 70% at 50 mM concentration. Paclitaxel (PTX; 6 mg/mL) and Gemcitabine (GEM; 50 µM) were purchased by Sigma Aldrich and solubilized in water. All the compounds were aliquoted and stored at −20 °C and each aliquot was used one time.

### 4.5. O_2_/O_3_ Treatments

PANC-1 and MiaPaCa-2 cell lines were seeded on 96-well culture plates, or 12-well plates at a density of 3.0 × 10^4^ cells/mL. The cells were pre-cultured in normoxia for 24 h. Subsequently, the culture plates were exposed twice to O_2_/O_3_ treatments for 30 min in a Hypoxia Incubator Chamber (Stemcell Technology, Vancouver, BC, Canada), by injecting O_2_/O_3_ until chamber saturation, using a Midi Ozon Active machine (Medica s.r.l., Bologna, Italy).

After treatment, the cell plates were replaced in the incubator at normoxia condition (37 °C with 5% CO_2_ and 95% humidity) for 6 h and then, the O_2_/O_3_ treatment was repeated as described. After that, the cells plates were maintained in normoxia for 24 up to 72 h, before performing the experiments.

### 4.6. MTT Assay

3 × 10^4^ cells/mL were seeded in 96-well plates in a final volume of 100 μL/well. After one day of incubation, compounds or vehicles, alone or in combination, were added and six replicates were used for each treatment and all experiments were repeated three times. In some experiments, the treatment was daily added, after washing with fresh medium. After 24 or 72 h cell viability was investigated by adding 0.8 mg/mL of 3-[4,5-dimethylthiazol-2-yl]-2,5 diphenyl tetrazolium bromide (MTT) (Sigma Aldrich) to the media. After 3 h the supernatant was removed, and the pellet of salt crystals was solubilized with 100 μL/well of DMSO. The absorbance of the sample against a background control was measured at 570 nm using an ELISA reader microliter plate (BioTek Instruments, Winooski, VT, USA).

### 4.7. Cell Cycle Analysis

Cells were seeded into 6-well plates (3 × 10^4^ cells/mL) and treated with CBD (12.5 μM), in the presence and absence of O_2_/O_3_, for 12 h. Cells were fixed by adding ice-cold 70% ethanol for 1 h and then washed with staining buffer (PBS, 2% FBS and 0.01% NaN3). Next, cells were incubated with 100 μg/mL Ribonuclease A solution (Sigma Aldrich) for 30 min at 37 °C, stained with propidium iodide (PI) 20 μg/mL (Sigma Aldrich) at room temperature for 10 min and analysed on a FACScan flow cytometer using linear amplification and CellQuest software, version 3.0 (BD Biosciences, San Jose, CA, USA).

### 4.8. Cell Death Assay

Annexin V-FITC and PI staining followed by FACS analysis was used to evaluate cell death on PANC-1 and MiaPaCa-2 cancer cell lines. Cells at a density of 3 × 10^4^ cells/mL were treated with different doses of CBD, in the presence and absence of O_2_/O_3_, for 24 up to 48 h. After treatment, cells were stained with 5 μL of Annexin V-FITC or with 20 μg/mL PI for 10 min at room temperature and then washed with binding buffer (10 mM N-(2-Hydroxyethyl)piperazine-N0-2-ethanesulfonic acid [HEPES]/sodium hydroxide, pH 7.4, 140 μM NaCl, 2.5 μM CaCl_2_). The percentage of positive cells determined over 10,000 events was analysed with FACScan flow cytometer using the CellQuest software. All experiments were repeated three times.

### 4.9. Alkaline Comet Assay

Cell lines were plated in 6-well plates (3 × 10^4^ cells/mL) one day before treatment exposure. Semi-confluent cultures were daily treated for up to 48 h with CBD 25 µM or vehicle. Cells treated with vehicle were included. All experiments were repeated three times. The comet assay was performed under alkaline conditions following the ABCAM protocol. Briefly, after exposure to treatments, the cells were resuspended in 1 × PBS and added to 75 µL of molten (37 °C) 0.5% low-melting-point agarose gel to achieve a cell concentration of 1 × 10^5^ cells/mL. The agarose was pipetted onto the comet slides. Slides were stored in the dark at 4 °C for 10 min before adding pre-chilled lysis buffer for 45 min at 4 °C in the dark. The slides were immersed in freshly prepared alkaline solution (0.25 M NaOH containing 0.1 µM EDTA, pH 12.6) for 30 min at the same conditions. Slides were then removed and washed twice with TBE buffer for 5 min. Gel electrophoresis was performed at 1 V cm^−1^ (length of the electrophoretic chamber) for 20 min (running amperage 3–5 mA with the distance between the two electrodes of 25 cm). The comet slides were washed with 70% ethanol for 5 min and air-dried for 1 h at room temperature. 100 µL of diluted SYBR Green solution was placed onto each dried agarose circle. The slides were then read with a fluorescence microscope (Leica Microsystems, Buccinasco, Italy).

### 4.10. Wound-Healing Assay

PANC-1 and MiaPaCa-2 cells were seeded onto a 24-well plate at a density of 4 × 10^4^/mL and 2.5 × 10^4^/mL, respectively. When cells were confluent, they were scratched using 10 μL sterile pipette tips, and medium was replaced with fresh medium containing low percentage of serum to minimize cell proliferation and prevent cell detachment and treated with CBD 12.5 μM. Images of wounded areas were taken at 0 h and 48 h. Images acquisition was carried out by a LeitzFluovert FU (Leica Microsystems) microscope. Remaining wound areas were determined using NIH Image J 1.44 software (Research Services Branch (RSB), National Institutes of Health (NIH), Bethesda, MD, USA) three separate distance measurements per well. Analyses were performed in triplicate.

### 4.11. Statistical Analysis

The data presented represent the mean with standard deviation (SD) of at least 3 independent experiments. Synergistic activity of the CBD and chemotherapeutic drugs combination was calculated by the Chou-Talalay method, which provides the theoretical basis for the combination index (CI)-isobologram equation. This method allows quantitative determination of drug interactions, where CI < 1, = 1, and > 1 indicates synergism, additive effect and antagonism, respectively. Based on these algorithms, computer software, CompuSyn 3.0.1 version (CompuSyn Software, ComboSyn, Inc., Paramus, NJ, USA, 2007) was used for automatically determining synergism and antagonism at all doses or effect levels [66]. The statistical significance was determined by Student’s *t*-test and by One Way-Anova and Two Way-Anova with Bonferroni’s post-test; * *p* < 0.05.

## 5. Conclusions

In conclusion, we found that CBD and O_2_/O_3_ were both able to induce significant changes in the expression profile of genes strongly involved in PDAC, suggesting that further study on this combination could be addressed to better elucidate the role of CBD and O_2_/O_3_ in the progression of PDAC.

## Figures and Tables

**Figure 1 cancers-12-02774-f001:**
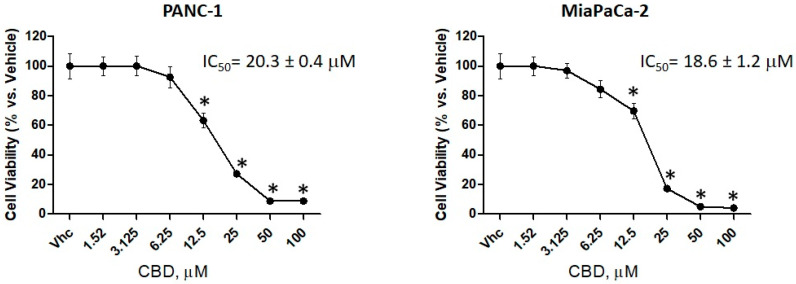
Cannabidiol (CBD) induced cytotoxicity in PDAC cell lines. Cell viability was determined by 3-[4,5-dimethylthiazol-2-yl]-2,5 diphenyl tetrazolium bromide (MTT) assay. PANC-1 and MiaPaCa-2 cells were treated for 72 h with different concentrations of CBD (up to 100 µM), in daily administration. Data shown are expressed as mean ± SE of three separate experiments. * *p* < 0.05 treated vs. vehicle.

**Figure 2 cancers-12-02774-f002:**
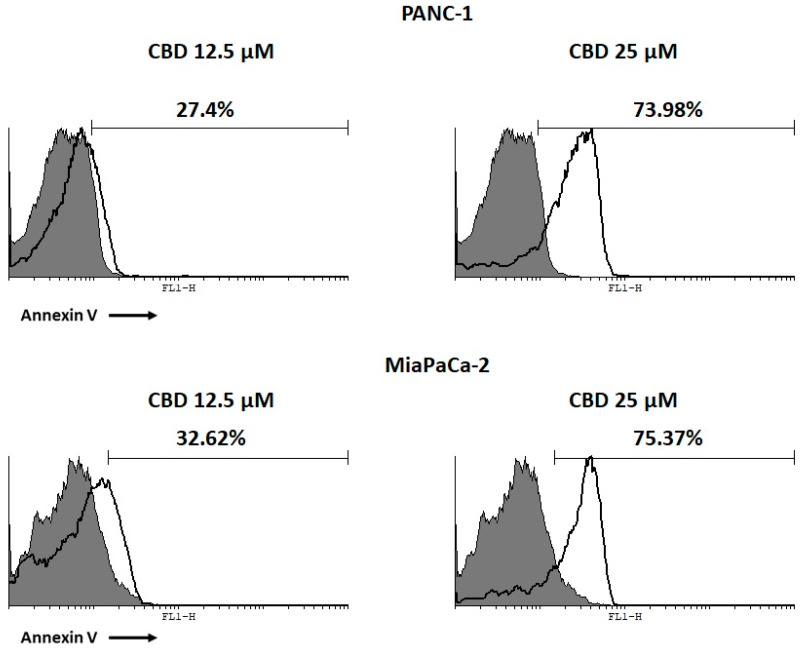
CBD induced cell death in pancreatic ductal adenocarcinoma (PDAC) cell lines. PDAC cell lines were treated with CBD for 48 h. Flow cytometric analysis was performed by Annexin V/Propidium Iodide (PI) staining. Data represent the percentage of Annexin V positive cells and are representative of one of three separate experiments.

**Figure 3 cancers-12-02774-f003:**
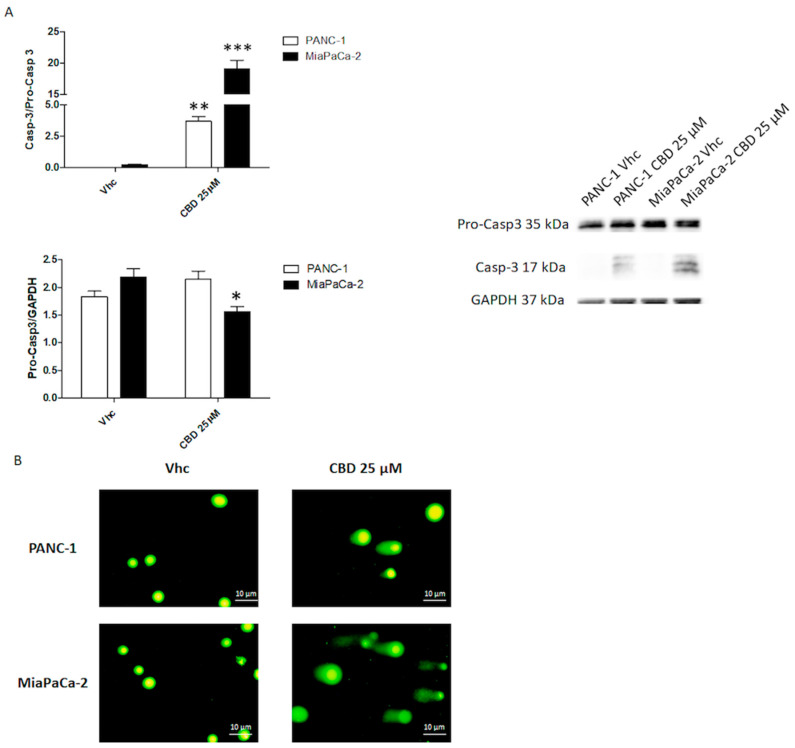
CBD induced apoptotic cell death in PDAC cell lines. PDAC cell lines were treated with CBD for 48 h. (**A**) Western blot analysis and densitometric quantification of Casp-3 protein levels. Pro-Casp3 densitometric values were normalized to Glyceraldehyde 3-phosphate dehydrogenase (GAPDH) used as loading control, Casp-3 densitometric values were normalized to Pro-casp3. Blots are representative of one of three separate experiments, * *p* < 0.05, ** *p* < 0.01, *** *p* < 0.001 treated vs. untreated cells. The whole western blot image can be found in Appendix A. (**B**) Cell damage and DNA fragmentation were determined on PANC-1 and MiaPaCa-2 cells untreated (Vehicle Vhc) and treated with CBD for 48 h by Comet assay (alkaline electrophoresis conditions 20 V for 10 min, image acquisition 10×).

**Figure 4 cancers-12-02774-f004:**
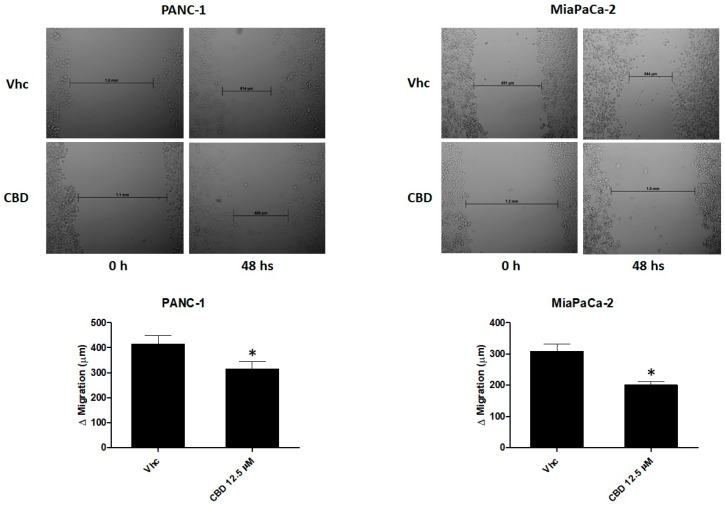
CBD treatment inhibits migration of PDAC cells. Representative image of wound-healing assays for PANC-1 and MiaPaCa-2 cells after treatment with CBD 12.5 µM for up 48 h. All experiments were repeated three times and images were taken at 0 and 48 h (10×). Data are presented as the mean ± SE. * *p* < 0.05 vs. Vhc.

**Figure 5 cancers-12-02774-f005:**
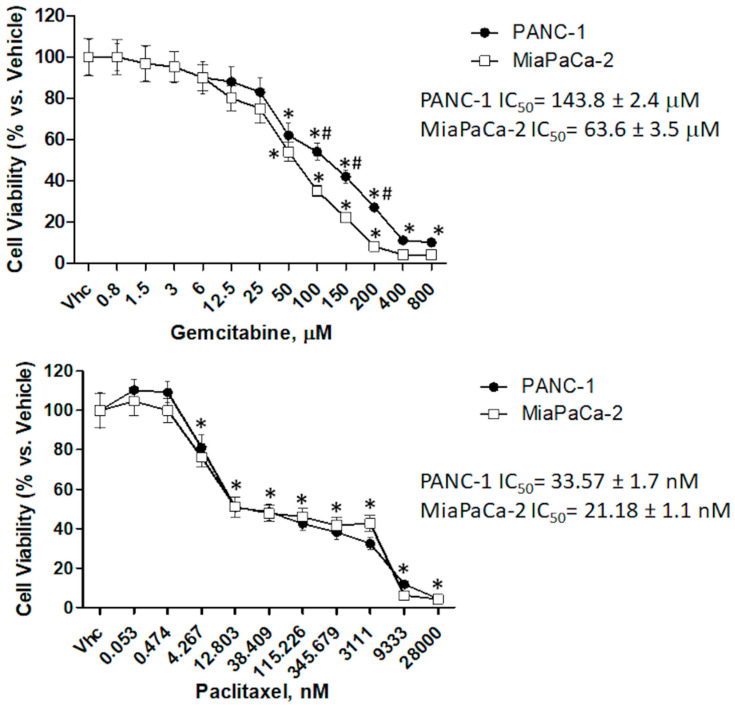
Cytotoxic effect of chemotherapeutic drugs in PDAC cell lines. Cell viability was determined by MTT assay. PANC-1 and MiaPaCa-2 cells were treated for 72 h with different concentrations of gemcitabine (GEM) (up to 800 µM) or paclitaxel (PTX) (up to 28 µM), in single administration. Data shown are expressed as mean ± SE of three separate experiments. * *p* < 0.05 treated vs. vehicle, ^#^
*p* < 0.05 PANC-1 vs. MiaPaCa-2.

**Figure 6 cancers-12-02774-f006:**
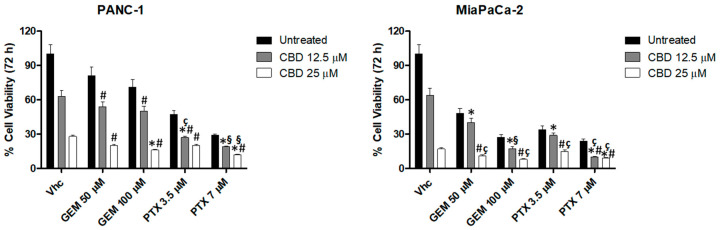
CBD increases the cytotoxic effect of chemotherapeutic drugs in PDAC cell lines. CBD increases the cytotoxic effect of chemotherapeutic drugs in PDAC cell lines. Cell viability was determined in PDAC cell lines by MTT assay. Cells were treated for 72 h with CBD, alone and in combination with different doses of GEM and PTX. Data shown are expressed as mean ± SE of three separate experiments. * *p* < 0.05 vs. CBD alone, ^#^
*p* < 0.05 vs. chemotherapeutic drug alone, ^§^ for additive effect, ^ç^ for synergistic effect.

**Figure 7 cancers-12-02774-f007:**
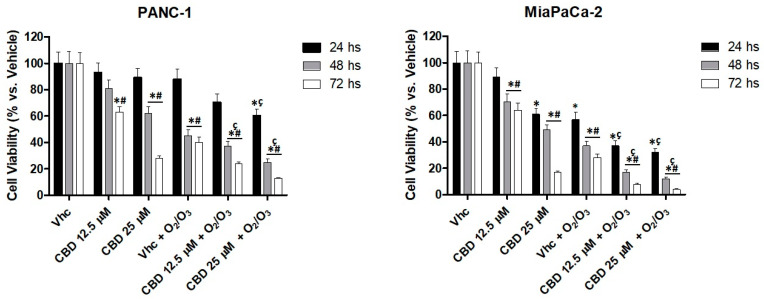
O_2_/O_3_ improved CBD cytotoxicity in PDAC cell lines. Cell viability, up to 72 h, was determined by MTT assay. PANC-1 and MiaPaCa-2 cells were treated with O_2_/O_3_ and CBD (12.5 and 25 µM) in daily administration. Data shown are expressed as mean ± SE of three separate experiments. * *p* < 0.05 treated vs. vehicle, ^#^
*p* < 0.05 48, 72 h vs. 24 h, ^ç^
*p* < 0.05 vs. CBD.

**Figure 8 cancers-12-02774-f008:**
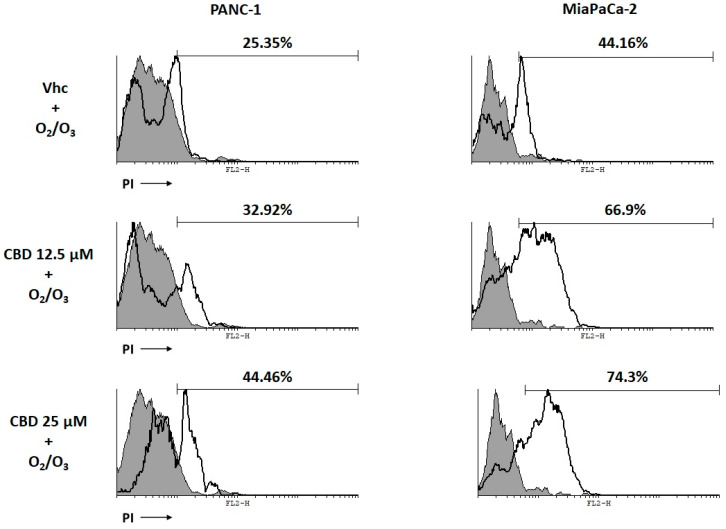
O_2_/O_3_ induced necrotic cell death in PDAC cell lines. PDAC cell lines were treated with O_2_/O_3_ and CBD for 24 h. Flow cytometric analysis was performed by Annexin V/PI staining. Data represent the percentage of PI positive cells and are representative of one of three separate experiments.

**Figure 9 cancers-12-02774-f009:**
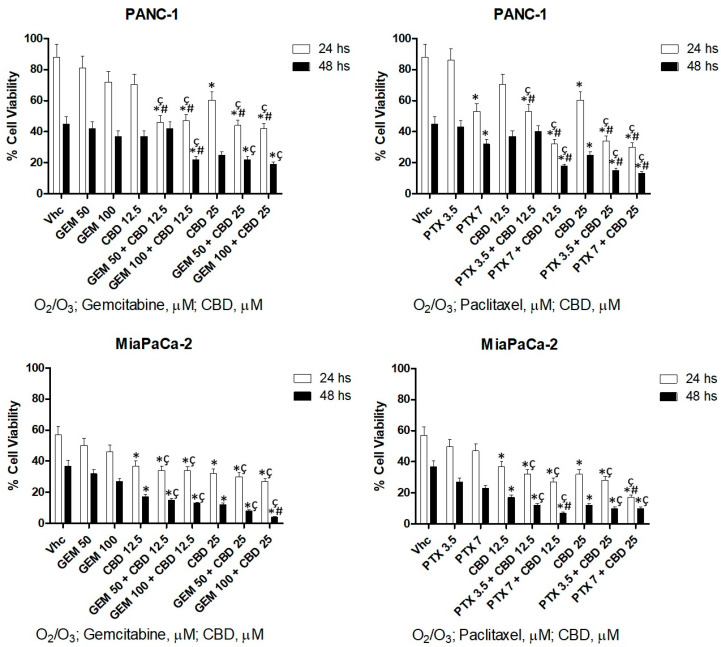
Addition of O_2_/O_3_ improves cytotoxic effect of CBD and chemotherapeutic drugs. Cell viability was determined in PDAC cell lines by MTT assay, and the percentage of cell viability was calculated compared with O_2_/O_3_ alone. Vhc showed in the graph is O_2_/O_3_ alone. O_2_/O_3_ conditioned cells were treated for 24 and 48 h with CBD, alone and in combination with different doses of GEM and PTX. Data shown are expressed as mean ± SE of three separate experiments. * *p* < 0.05 vs. Vhc ^#^
*p* < 0.05 vs. CBD alone, ^ç^
*p* < 0.05 vs. chemotherapeutic drug alone.

**Figure 10 cancers-12-02774-f010:**
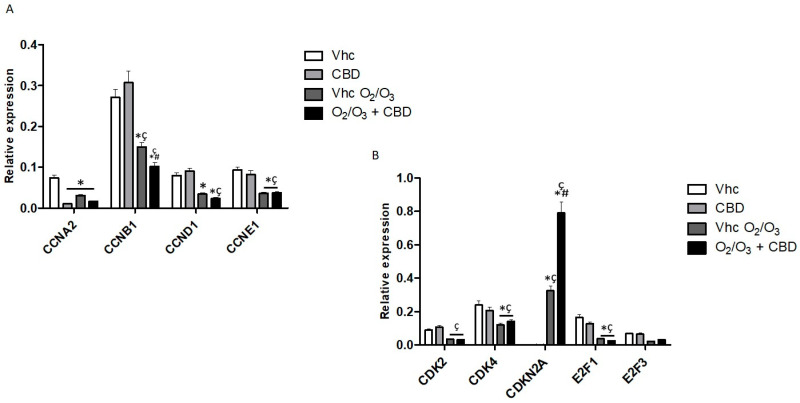
Modulation of cell cycle pathways in PANC-1 cell line. (**A**) *CCNA2*, *CCNB1*, and *CCND1*, *CCNE1* mRNA expression and (**B**) *CDK2*, *CDK4*, *CDKN2A*, *E2F1*, and *E2F3* mRNA expression was evaluated by qRT-PCR in the PANC-1 cell line, treated with CBD in the presence and absence of O_2_/O_3_. Target mRNA levels were normalized for *GAPDH* expression. Data are expressed as fold mean ± SE. * *p* < 0.05 vs. Vhc, ^#,^* *p* < 0.05 vs. Vhc O_2_/O_3_, ^ç^
*p* < 0.05 vs. CBD.

**Figure 11 cancers-12-02774-f011:**
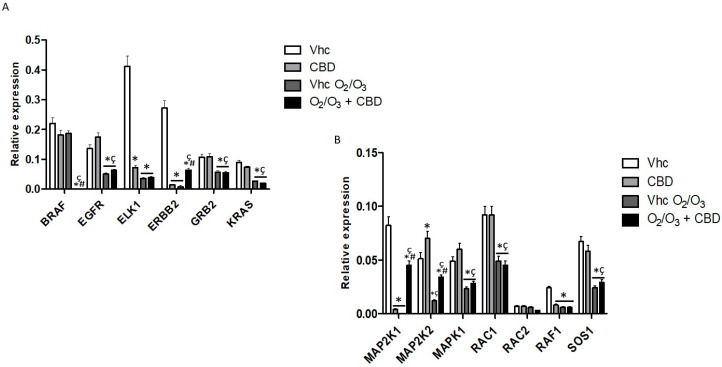
Modulation of Ras pathways in PANC-1 cell line. (**A**) *BRAF*, *EGFR*, *ELK1*, *ERBB2*, *GRB2*, and *KRAS* mRNA expression and (**B**) *MAP2K1*, *MAP2K2*, *MAPK1*, *RAC1*, *RAC2*, *RAF1*, and *SOS1* mRNA expression was evaluated by qRT-PCR in the PANC-1 cell line, treated with CBD in the presence and absence of O_2_/O_3_. Target mRNA levels were normalized for *GAPDH* expression. Data are expressed as fold mean ± SE. * *p* < 0.05 vs. Vhc, ^#,^* *p* < 0.05 vs. Vhc O_2_/O_3_, ^ç^
*p* < 0.05 vs. CBD.

**Figure 12 cancers-12-02774-f012:**
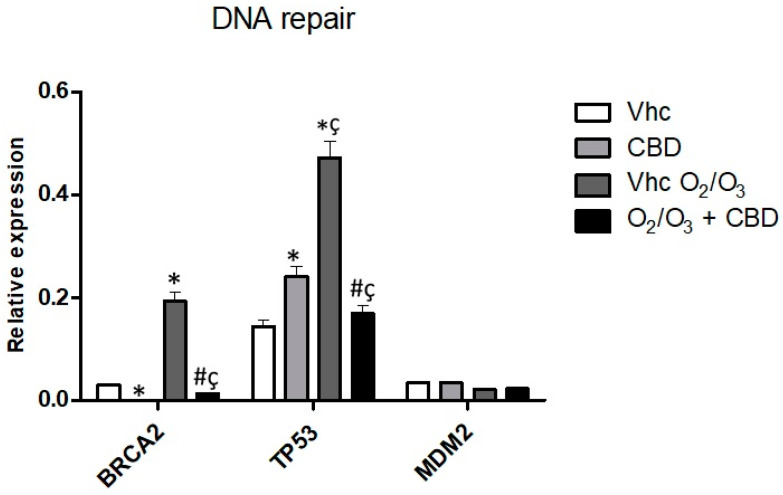
Modulation of DNA repair pathways in PANC-1 cell line. *BRCA2*, *TP53*, and *MDM2* mRNA expression was evaluated by qRT-PCR in the PANC-1 cell line, treated with CBD in the presence and absence of O_2_/O_3_. Target mRNA levels were normalized for *GAPDH* expression. Data are expressed as fold mean ± SE. * *p* < 0.05 vs. Vhc, ^#,^* *p* < 0.05 vs. Vhc O_2_/O_3_, ^ç^
*p* < 0.05 vs. CBD.

**Figure 13 cancers-12-02774-f013:**
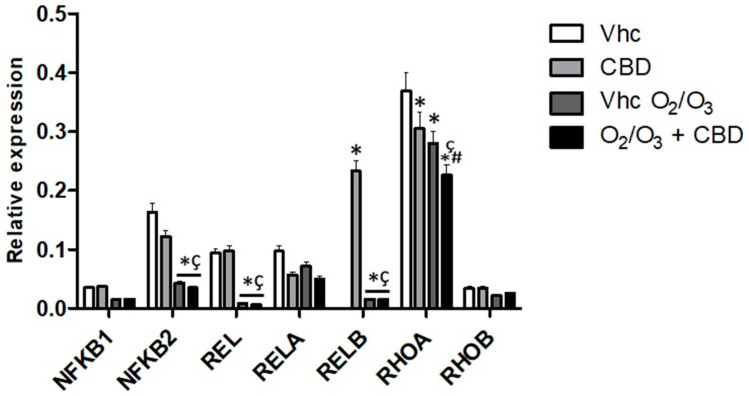
Modulation of NF-kB pathways in PANC-1 cell line. *NFKB1*, *NFKB2*, *REL*, *RELA*, *RELB*, *RHOA*, and *RHOB* mRNA expression was evaluated by qRT-PCR in the PANC-1 cell line, treated with CBD in the presence and absence of O_2_/O_3_. Target mRNA levels were normalized for *GAPDH* expression. Data are expressed as fold mean ± SE. * *p* < 0.05 vs. Vhc, ^#,^* *p* < 0.05 vs. Vhc O_2_/O_3_, ^ç^
*p* < 0.05 vs. CBD.

**Figure 14 cancers-12-02774-f014:**
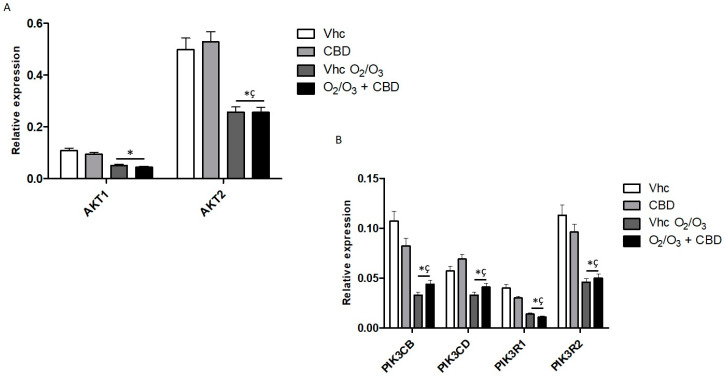
Modulation of PI3K/AKT pathway in PANC-1 cell line. (**A**) *AKT1* and *AKT2* mRNA expression and (**B**) *PIK3CB*, *PIK3CD*, *PIK3R1* and *PIK3R2* mRNA expression was evaluated by qRT-PCR in PANC-1 cell line, treated with CBD in presence and absence of O_2_/O_3_. Target mRNA levels were normalized for *GAPDH* expression. Data are expressed as fold mean ± SE. * *p* < 0.05 vs. Vhc, ^#,^* *p* < 0.05 vs. Vhc O_2_/O_3_, ^ç^
*p* < 0.05 vs. CBD.

**Table 1 cancers-12-02774-t001:** IC_50_ values and percentages of viability in PANC-1, MiaPaCa-2, H6c7 and NHF cell lines with CBD and O_2_/O_3_, alone or in combination, expressed as mean ± SD of three separate experiments.

Treatment	PANC-1	MiaPaCa-2	H6c7	NHF
CBD IC_50_ (µM)	20.3 ± 0.4	18.6 ± 1.2	28.6 ± 0.6	30.6 ± 1.1
Vhc O_2_/O_3_ % of viability (72 h)	40.2 ± 3.8	28.1 ± 2.7	92.2 ± 8.1	89.1 ± 5.7
CBD 12.5 µM + O_2_/O_3_ % of viability (72 h)	24.1 ± 1.1	8.1 ± 0.7	87.9 ± 6.5	84.5 ± 2.6
CBD 25 µM + O_2_/O_3_ % of viability (72 h)	13.1 ± 0.4	4.1 ± 0.4	50.9 ± 4.1	54.6 ± 3.2

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
