# Peer review of "Cannabidiol and Oxygen-Ozone Combination Induce Cytotoxicity in Human Pancreatic Ductal Adenocarcinoma Cell Lines"

_cancers, 2020, doi:10.3390/cancers12102774_

Round 1

Reviewer 1 Report

In this manuscript Luongo et al. study the antitumoral effect of combined therapy with Cannabidiol and oxygen-ozone therapy in two PDAC cell lines. This field is of interest, however a major limitation regarding the research design must be pointed: It will be mandatory the inclusion of, at least, one of the availables normal pancreatic cell line (eg. H6c7) to consider the obtained results interesting and of relevance.

Reviewer 2 Report

Overall, the manuscript is well-written, logically presented e gives a  evidenced antitumoral effect of combined therapy with CBD and oxygen-ozone therapy in PDAC.

It would be interesting to evaluate the effect of selective blockers to CBD targets to clarify the involvement of CB1, CB2, GPR55, transient potential receptors or PPAR.  The authors could perform a series of experiments using selective antagonists (where available).

  1. Abbreviations: some are not in agreement
  2. List of abbreviations are missing

Reviewer 3 Report

This study aims to assess, in-vitro the effect of CBD treatment and/or Ozone therapy with/out combination with anti-cancer drugs in the treatment of PDAC. 

Comments: 

Introduction:

Please expand on the hypotheses leading to testing the combination of CBD and O3/O2 therapy?

Methods:

Can you please specify how many biological repeats/technical repeats were performed?

MTT Assay - Is this the best method to assess cell viability? MTT is a metabolic, NADH dependent test. Ozone therapy increase redox stress and therefore might interfere with interpretation of the results.

qPCR analysis was based on normalization to GAPDH mRNA levels - Have the authors verified GAPDH mRNA levels are a good control for normalization? GAPDH expression has been shown to change upon oxidative stress.

Results:

Figure 2.

Annexin V is a rather early apoptotic marker. Was a co-stain with PI also performed? Were any positive and negative controls used? how many times was this experiment performed? 

Figure 3.

Is GAPDH a good normalization control? Were any positive and negative controls used?

Figure 4.

Was migration assessed by measuring distance or area? if it was distance based, how many separate distance measurements were performed per well? Were positive and negative controls also assessed?

Figure 5.

The IC50 values are considerably higher from previously published data. Usually GEM and PTX IC50s were described to be in the low nanomolar range. How was the MTT performed? is there a chance the wells became overconfluent which skewed the data?

Figure 6.

The correct way to assess whether a combination is additive,synergistic or antagonistic is with a CI analysis or  BLISS/LOEWE plots. The data presented does not necessarily support a synergistic or additive effect and might in fact be antigonistic (as the MiaPaca2 PTX combination suggests).

Figure 7.

Please see my comments regarding MTT assays. Is there a viability assay done for O2-O3 therapy only? The effects of that should be shown prior to combinations.  

Figure 8.

Why have the researchers chosen to use PI staining for the evalulation of ozone therapy and Annexin V for CBD therapy?  Why not use both? Why the different stains? Were any positive/negative controls assessed?

Figure 10.

The conclusion of "Summarising, these data strongly support that the
231 co-treatments with O2/O3 and CBD was efficacy in reducing cell cycle progression in PDAC cell lines" is overstated. At the very least, some form of cell cycle analysis (e.g BrdU/EdU staining combined with PI) is warranted in order to comment on the effects on cell cycle.

Round 2

Reviewer 1 Report

All my concerns have been adressed. The inclusion of the recommended new cell lines, together with the news HKGs included, represents a significant improvement in the study. I recommend the manuscript for publication in the present form. 

Author Response

Thanks for the comments.

Reviewer 3 Report

The authors have addressed several of the concerns in a satisfactory way. However, some major concerns still remain:

MTT study design - Miapaca2 cells are very fast replicating. Plating 30,000 cells per well in a 96-well plate will usually result in confluence within 2 days.

Figure 6 / CBD combination data with GEM or PTX : The title is misleading "CBD increases the cytotoxic effect of chemotherapeutic drugs in PDAC cell lines" as the CI data actually suggests that most combination resulted in antagonism. The results of the CI analysis should stated clearly and added to the supplementary data.

Figure 7 / Same as with figure 6 - Data should be analyzed in a similar fashion. Currently plots are actually suggestive of some antagonism with addition of O2/O3 in the Panc1 cells and synergic/additive in the Miapaca2.

Figure 9/ The description states that "Cell viability was calculated comparing with untreated cells in presence of O2/O3", However some of the graphs show the Vhc cells (which I assume are basically untreated) as having 80-60% viability. Please explain what was defined as 100%?

Gene expression data: Care needs to be taken when interpreting data from a large gene panel. Were FDR calculation employed? Was this screen validated with protein/IHC data?

In order to suggest a change to cell cycle progression , Please add another validative cell cycle test (test such as EdU/BrdU & PI flow cytometry).

minor comments:

Paclitaxel cell viability graph does not reach a full sigmoid  - lower concentrations need to be assessed as well.
